# Characterization of RAGE and CK2 Expressions in Human Fetal Membranes

**DOI:** 10.3390/ijms24044074

**Published:** 2023-02-17

**Authors:** Karen Coste, Shaam Bruet, Caroline Chollat-Namy, Odile Filhol, Claude Cochet, Denis Gallot, Geoffroy Marceau, Loïc Blanchon, Vincent Sapin, Corinne Belville

**Affiliations:** 1iGReD, Team “Translational Approach to Epithelial Injury and Repair”, UMR6293 CNRS-U1103 INSERM, Université Clermont Auvergne, F-63000 Clermont-Ferrand, France; 2CHU Clermont-Ferrand, Neonatal Intensive Care Department, F-63000 Clermont-Ferrand, France; 3INSERM, CEA, UMR Biosanté, U1292, University Grenoble Alpes, F-38000 Grenoble, France; 4CHU Clermont-Ferrand, Obstetrics and Gynecology Department, F-63000 Clermont-Ferrand, France; 5CHU Clermont-Ferrand, Biochemistry and Molecular Genetic Department, F-63000 Clermont-Ferrand, France

**Keywords:** fetal membranes, amniocyte, RAGE, CK2, rupture, labor, inflammation

## Abstract

At the feto-maternal interface, fetal membranes (FM) play a crucial role throughout pregnancy. FM rupture at term implicates different sterile inflammation mechanisms including pathways activated by the transmembrane glycoprotein receptor for advanced glycation end-products (RAGE) belonging to the immunoglobulin superfamily. As the protein kinase CK2 is also implicated in the inflammation process, we aimed to characterize the expressions of RAGE and the protein kinase CK2 as a candidate regulator of RAGE expression. The amnion and choriodecidua were collected from FM explants and/or primary amniotic epithelial cells throughout pregnancy and at term in spontaneous labor (TIL) or term without labor (TNL). The mRNA and protein expressions of RAGE and the CK2α, CK2α′, and CK2β subunits were investigated using reverse transcription quantitative polymerase chain reaction and Western blot assays. Their cellular localizations were determined with microscopic analyses, and the CK2 activity level was measured. RAGE and the CK2α, CK2α′, and CK2β subunits were expressed in both FM layers throughout pregnancy. At term, RAGE was overexpressed in the amnion from the TNL samples, whereas the CK2 subunits were expressed at the same level in the different groups (amnion/choriodecidua/amniocytes, TIL/TNL), without modification of the CK2 activity level and immunolocalization. This work paves the way for future experiments regarding the regulation of RAGE expression by CK2 phosphorylation.

## 1. Introduction

Fetal membranes (FMs) enclose the fetus during the nine months of pregnancy. This transitional structure is made of two layers: the amnion, which is in contact with amniotic fluid, and the chorion, which is based on the maternal decidua. Even if these layers are extra-embryonic tissues, they play important roles during pregnancy and parturition. In fact, they protect the fetus, growing and maturing in tandem with it. A few weeks before term delivery, the FM will gradually weaken and dissociate, causing the chorion—and subsequently the amnion—to rupture for the birth of the baby (after 37 weeks of gestation) [1]. This physiological tearing of the FM occurs in a specific area overlying the cervix called the “zone of altered morphology” (ZAM), which is subjected to remodeling by a multitude of signals that are mechanical, biochemical, or cellular [2]. FM will gradually prepare to rupture at the end of pregnancy through a mechanism of senescence, or an aging process, in which senescent membranes display an inflammatory profile described as a senescence-associated secretory phenotype [3,4]. In direct linkage with this mechanism are molecules named alarmins or damage-associated molecular patterns (DAMPs) such as the cell-free DNA, the S100 protein family, and high-mobility group box 1 (HMGB1) proteins are released in the FM. They bind to pattern recognition receptors (PRRs) which leads to a microbial-free inflammatory response called sterile inflammation [5]. When all of these physiological phenomena are abnormally amplified, DAMPs and PPRs are overexpressed in maternal serum or the FM [6,7], promoting an epithelial-to-mesenchymal transition (EMT) [8,9,10]. This can lead to premature tissue remodeling and FM rupture before 37 weeks of gestation in 2–3% of pregnancies [11].

Recently, we demonstrated that at term without labor (TNL), the receptor for advanced glycation end-products (RAGE) belonging to the PPRs family and its ligand HMGB1 were overexpressed in the amnion compared with the choriodecidua at the site of ZAM rupture. The binding of HMGB1 to RAGE induced IL-1β and TNFα cytokine production in the FM, which led to a sterile inflammatory status [12]. RAGE, which belongs to the immunoglobulin superfamily of receptors, is also known to activate various intracellular signaling pathways and to cause inflammatory diseases such as diabetes, neurodegenerative diseases, or cancer. However, this transmembrane glycoprotein receptor does not have its own kinase activity to transduce its downstream cellular signal [13,14]. Thus, the manner in which its cytoplasmic domain is activated upon ligand binding and how its activation is transduced remain unknown. One hypothesis is that the phosphorylation of the cytoplasmic domain of RAGE on serine 391 by PKCζ increases its affinity for TIRAP and MyD88 adaptors, facilitating downstream cellular message transduction [15]. In pregnancy-related tissues such as the placenta, another kinase, protein kinase CK2, was recently described as being directly implicated in the control of the proliferation, migration, and invasion of trophoblast cells—a key phenomenon for a normal pregnancy [16]. This serine/threonine kinase is a constitutively active enzyme and has more than 300 molecules as substrates in eukaryotic cells [17]. It is a tetrameric holoenzyme formed by the association of two catalytic α and/or α′ subunits and two regulatory β subunits. CK2 is involved in many diseases, including local inflammatory induction [18], and actively participates in various cellular processes such as cell migration or EMT induction [19,20].

As CK2 has never been characterized in FM throughout the pregnancy and as personal results reveal that RAGE owns potential CK2 phosphorylation sites, we decide to describe the RAGE and CK2 expression profiles.

In this study, we demonstrate that CK2 is expressed in both FM and primary amniocytes during pregnancy, at full term after spontaneous labor (TIL) and after birth by cesarean section without labor (TNL) and that its cellular localization is similar to the RAGE expression profile. Although the data regarding the regulation of RAGE during FM rupture at term are scarce, they suggest that CK2 expression could be an indirect key effector of sterile inflammation by regulating the RAGE transduction level.

## 2. Results

### 2.1. Ontogeny of RAGE and CK2 Subunit mRNA Expressions in Fetal Membranes during the Three Trimesters of Human Pregnancy

To establish a longitudinal ontogeny of the α, α′, β CK2 subunits and RAGE mRNA expression throughout gestation, FM samples were collected in the first trimester (T1), the second trimester (T2), and the third trimester (T3) after labor. The mRNA expressions of the CK2α, CK2α′, and CK2β subunits and RAGE in the amnion and choriodecidua were quantified using the reverse transcription quantitative polymerase chain reaction (RT-qPCR) at the second (T2) and third (T3) trimesters of pregnancy and compared with those at the first trimester (T1), when the two tissues appear as one. The RAGE mRNA expression level in the amnion significantly increased between T1 and T2 and stabilized at T3. In the choriodecidua, the expression profile was slightly different, reaching a maximum level at T2 and decreasing at T3 until its T1 level (Figure 1A). At T2, the RAGE mRNA expression level was not statistically different between the amnion and choriodecidua, demonstrating that during this trimester, RAGE is expressed at the same level in both tissue layers.

During pregnancy, the expression level of the CK2α subunit increased substantially only in the amnion (Figure 1B), whereas the CK2α′ and CK2β mRNA levels were statistically constant throughout pregnancy, regardless of the FM layers (Figure 1C,D). However, the CK2β expression level was approximately twice as high as those of CK2α and CK2α′.

### 2.2. RAGE and CK2 Immunolocalizations in Fetal Membranes at Term in Case of Spontaneous Labor (TIL) and without Labor (TNL)

In the TNL group, cells in the amnion (epithelial amniocytes and fibroblasts cells) and choriodecidua (fibroblasts, trophoblasts, and decidual cells) expressed RAGE. However, the immunofluorescence signal intensity was particularly low in the FM in the TIL group (Figure 2, Appendix A).

At term, even if precise quantification using immunofluorescence staining was not adequate, the CK2α, CK2α′, and CK2β subunit protein expressions exhibited the same pattern as the RAGE expression in the FM with different signal intensities (Figure 2).

### 2.3. Molecular Characterization of RAGE and CK2 Subunits in Human Fetal Membranes at Term with Spontaneous Labor (TIL) and No Labor (TNL)

As mentioned earlier (Figure 1A), in the FM at TIL, the RAGE mRNA expression level at T3 was particularly high in the amnion compared with the choriodecidua (Figure 3, left). This difference was not observed at the protein expression level. By contrast, the RAGE protein level was specifically upregulated in the amnion in the TNL group compared with the TIL group (Figure 3, right). No significant difference in RAGE mRNA or protein expression level in the choriodecidua was found between the two groups (Figure 3). In the TIL group, the CK2α′ mRNA was the isoform clearly overexpressed in the amnion compared with the choriodecidua (Figure 4A); only a tendency was observed for CK2α. The CK2β mRNA expression level was higher in the amnion layer in the TIL group than in the TNL group (Figure 4A). No statistically significant differences were observed between the groups in terms of protein expression level in spite of the remarkable decrease in CK2α expression level compared with the CK2α′ and CK2β expression levels (Figure 4B). Finally, no significant difference in CK2 kinase activity level was observed in relation with factors such as the presence or absence of labor and the nature of the tissue layer (Figure 4C).

### 2.4. RAGE and CK2 Immunolocalizations in Primary Amniotic Epithelial Cells at Term in Case of Spontaneous Labor (TIL) and without Labor (TNL)

The subcellular localization of RAGE was cytosolic and was found at the cell membranes of the primary amniocytes, but the signal intensity was different at TIL and TNL. In the TIL group, RAGE labelling was much weaker than in the TNL group. In addition, RAGE was expressed at the membranes of the amniotic epithelial cells from the TNL group (Figure 5). The expression levels of the CK2 subunits were higher in the TIL group than in the TNL group. Their localizations varied depending on the subunit considered and independently of the tissue origin of the primary amniocytes. CK2α and CK2α′ were essentially expressed in the cytoplasm and to a lower extent in the nucleus, whereas CK2β was predominantly localized in the nucleus. Even if the signal quantification was difficult to analyze using immunofluorescence, CK2α′ and CK2β appeared to show greater signal intensity than CK2α (Figure 5, Appendix A).

### 2.5. Molecular Characterization of RAGE and CK2 Subunits in Primary Amniocytes at Term in Case of Spontaneous Labor (TIL) and without Labor (TNL)

The RAGE mRNA expression level in the primary amniocytes increased in the TIL group compared with the TNL group (Figure 6, left). However, after Western blot analysis, this difference was not observed at the protein expression level (Figure 6, right). The mRNA expression levels of RAGE and the CK2 subunits and the CK2 activity were similar between the TIL and TNL groups (Figure 6 and Figure 7). The CK2α protein expression level in amniocytes could not be rigorously determined (Figure 7B).

## 3. Discussion

Even if FMs are transitory tissues, they are subject to tissue remodeling throughout pregnancy, depending on cellular processes, according to fetal growth. In recent years, studies have clearly shown that FM rupture in physiological conditions is the consequence of localized inflammation triggered by the release of alarmins (DAMPs) from damaged cells, implicating, for example, the membrane receptor RAGE [21,22,23]. In a previous study in our laboratory, RAGE expression was detected in the two layers of FMs, the amnion and choriodecidua, at term without labor (TNL) using reverse transcription polymerase chain reaction (RT-PCR) during the three trimesters of pregnancy. Furthermore, at term, RAGE was overexpressed in the amnion, particularly in the ZAM, a tissue in contact with alarmins in the amniotic fluid [12]. However, an important gap remains in understanding how RAGE expression could be regulated after the binding of its various ligands, particularly during the FM rupture mechanism [6,24]. One potential explanation is that CK2 is ubiquitously expressed and already described in organs implicated in parturition, such as the placenta, decidua, and cervix [16,25,26,27]. We hypothesized that this protein kinase, which is present in FMs, could influence the expression and/or activation of RAGE by phosphorylation. In this context, our work precisely describes the RAGE and CK2 ontogeny during pregnancy and at-term parturition with spontaneous labor (TIL) and without labor (TNL) in both FM and primary amniocyte samples. During pregnancy, at T2 and T3, RAGE mRNA was overexpressed in the amnion compared with the choriodecidua. CK2α and CK2α′ expressions were detected in the amnion and choriodecidua at the three trimesters. This result was unexpected because the CK2α′ isoform was found predominantly in the brain and testis, unlike the CK2α isoform, which had low cell and organ specificities [28]. We found that the expression levels of the CK2 subunits were independent of the FM layers and labor conditions.

The second part of this work was performed to determine the involvement of RAGE and CK2 in the FM at term pregnancy with spontaneous labor (TIL) or without labor (TNL), that is, before and after parturition. In the event of a caesarean delivery at term, FMs are removed surgically without prior labor (TNL) and are therefore intact. As such, they represent the state of senescence and EMT at the end of pregnancy, without the consequences of labor and placental expulsion. During at-term vaginal delivery, FMs are recovered after spontaneous labor (TIL) and ruptured. Thus, FMs display different molecular and cellular patterns at term without labor (TNL), undergoing other processes such as oxidative stress and increased inflammatory load [29]. Overall, our results show that the amnion and choriodecidua exhibit different RAGE mRNA expression profiles, with the highest expression level observed in the amnion. Moreover, in agreement with the findings of [12], the RAGE protein was overexpressed in the amnion of the without-labor (TNL) group compared with the spontaneous-labor (TIL) group, but this difference was not found in the choriodecidua. This could be partly explained by the constitution of the stock at the end of pregnancy, which can be “consumed” after activation during parturition. The mRNA expressions of the catalytic CK2α and CK2α′ subunits are similar but two times lower than that of the regulatory CK2β subunit throughout pregnancy and at term, suggesting the formation of different forms of CK2 holoenzymes (CK2 αα′β_2_, α_2_β_2_, and α′_2_β_2_). Nevertheless, we found that the CK2α protein expression level was significantly lower than those of the two other subunits in the FM and amniocytes, which suggests that the CK2 holoenzyme in the FM mainly consisted of CK2α′ and CK2β subunits. This is in accordance with the findings of previous studies that have shown the presence of a membrane-bound form of CK2 in epithelial cells [30], expanding the role of the kinase as a regulator of membrane proximal signals in neuroinflammatory disease [31].

The global sequence similarity between the two CK2 catalytic subunits is approximately 75%, and they differ in the C-terminal sequence protein; CK2α′ is 41 residues shorter than CK2α [32]. Moreover, the ATP-binding sites of the human CK2 catalytic subunits (between the N- and C-terminal lobes in a three-dimensional structure) differ by two amino acids (His115–Val116 for CK2α and Tyr116–Ile117 for CK2α′) [33]. Even though these structural differences have no effect on the kinetic constants, the affinity of CK2α′ for CK2β is approximately 12 times lower than its affinity for CK2α [34]. This feature may be important because the interaction between the CK2α/α′ and CK2β subunits is essential for substrate selectivity and live-cell imaging studies [30,35]. The unbalanced expressions of the CK2 subunits in various tumors suggest that CK2 subunits can coexist in the cell without forming an holoenzyme complex [19,20,36].

To our knowledge, this is the first time that the preponderant expression of CK2α′ is described in a female reproductive tissue. The degradation of CK2α mRNA could be a consequence of posttranscriptional regulation by miRNA. As already described by our team, let-7a-2 regulates the expression of the membrane receptor TLR4 and could preferentially target CK2α mRNA based on in silico analyses using the public database miRWalk [37]. Beyond the mRNA expression results in this study, we did not observe any statistical changes in the CK2 subunit protein expression by Western blot analyses, the CK2 kinase activity, or the modification of their cellular localizations, which suggests a constant CK2 expression level in the FM layers throughout pregnancy. In addition to its role in controlling the substrate specificity and cellular localization of the CK2 holoenzyme complex, CK2β possesses important functions in inhibiting the induction of EMT during cancer progression [19,20] and as a gatekeeper of focal adhesions in cell spreading [38].

As we did not observe any differences in the CK2β protein levels in the FM and amniocytes between the without-labor (TNL) and spontaneous labor (TIL) groups, we can hypothesize that the EMT sets up at term, well before FM rupture. RAGE, which has been described in tumor development [39] and cytoskeletal reorganization (Buckley et al., 2011) during EMT, could be involved in the EMT process and amplified in the TNL group owing to RAGE overexpression. Moreover, EMT may be correlated with the formation of “stress fibers” due to the accumulation of RAGE protein in the amniocytes from the without-labor (TNL) group, contributing to cell contraction and migration [8,10,40].

As we demonstrated that RAGE and CK2 are both present in the FM throughout pregnancy, we could postulate that CK2 could regulate the RAGE expression by phosphorylation and influence different physiological processes. To support this hypothesis, it has been shown that in HUVEC cells, in the presence of the RAGE ligand AGE, PKCζ phosphorylates serine 391 in the intracellular domain of endogenous RAGE. This leads to an increase in the receptor affinity for the two adaptive proteins TIRAP and MyD88, underlying the recruitment and activation of a cellular kinase and the induction of inflammatory cytokines via the transcription factor NF-κB [15,41]. Thus, it could be reasonably speculated that this type of cellular regulation could exist in the FM environment for the following reasons: (i) in silico analyses revealed that RAGE has several potential consensus CK2 phosphorylation sites in its intracellular domain with high software probability, (ii) several substrates for CK2 have been identified at the plasma membrane [42], and (iii) RAGE adaptors have been identified in FM and during sterile inflammation due to the synthesis of interleukins such as TNF-α and IL-1β [12]—one of the cellular processes contributing to FM rupture [43]. Moreover, CK2 is also involved in the placentation process, where its expression level is increased at the end of the first trimester, when oxygenation changes. Therefore, CK2 is also recognized as a key enzyme for the invasion of trophoblast cells, playing an essential role in future feto-maternal circulation. Moreover, increased CK2 activity due to an elevated CK2α/CK2β protein expression level has been shown to lead to the placental dysfunction responsible for preeclampsia [16].

Our study has several limitations as follows: (i) Our results relied on a small number of human FM samples, and (ii) because we had to make amniocyte pools to obtain enough material for protein quantification, we conducted our experiments on primary cells (physiological situation) rather than on a cell line. (iii) Herein, we focused on one cell type, amniocytes, which are in direct contact with amniotic liquid. In the future, this work could be completed by creating an in vitro feto-maternal interface model owing to the recent generation of stable cell lines of the FM (epithelial and mesenchymal cells from the amnion and chorion) and decidua parietalis cells, which are functionally similar to their primary cell counterparts [44].

In conclusion, we demonstrate that RAGE and CK2 are expressed in the two layers of FM throughout pregnancy and at the end of pregnancy with or without labor. We report that the RAGE expression level was higher at term without labor (TNL) than at term with spontaneous labor (TIL). We found that the CK2 in the FM was mainly composed of the catalytic CK2α′ and regulatory CK2β subunits. Further research must be conducted to demonstrate that RAGE could be phosphorylated by CK2 to regulate its cellular signaling regarding FM rupture, with a particular focus on well-described mechanisms such as EMT and sterile inflammation.

## 4. Materials and Methods

### 4.1. Human Fetal Membrane Collection

Fetal membranes were collected (between 14 March 2021 to 30 January 2022) from the Obstetrics Department, Estaing University Hospital, Clermont-Ferrand, France. All patients presented no underlying chronic or specific diseases of pregnancy (preeclampsia, pregnancy-induced hypertension, gestational diabetes, symptoms of preterm birth), no acute or chronic treatments (for hypertension, for example) as confirmed by (i) normal clinical following-up and (ii) macroscopic and microscopic placenta analyses and histological examinations that excluded chorioamnionitis. Furthermore, study patients (mother and baby) displayed no signs of sepsis ((French criteria: chorioamnionitis, urinary infections, isolated fever, elevated CRP, endometritis, early neonatal onset sepsis) during and after parturition; infection participation was rejected. Moreover, French clinical recommendations opted for the absence of amniotic fluid analyses for asymptomatic patients.

Samples of first-trimester (T1) membranes were obtained by aspiration after voluntary termination of pregnancy (10 to 12 weeks of gestation (WG)). Second-trimester (T2) membranes were collected after premature termination of pregnancy for severe fetal abnormalities (mainly severe cardiac malformations) without clinical repercussion for the mother’s health (22–25 WG). Macroscopic and microscopic analyzes were still normal. Third-trimester (T3) membranes were collected from pregnancies after spontaneous delivery (31–36 WG). Moreover, term FMs were collected from the women after spontaneous labor followed by vaginal deliveries (herein referred to as TIL, 37–41 WG) and after scheduled cesarean deliveries without labor due to scarred uterus and non-cephalic fetal position (herein referred to as TNL, 37–41 WG). After the collection of FM samples, they were immediately stored at −80 °C before quantitative RT-PCR, Western blot assay, or freezing sections. All membranes were separated into the amnion and choriodecidua (except for those collected during the first trimester), as previously described [45].

### 4.2. Cell Cultures

Primary amniocyte cells were collected from the living tissue of the amnion after trypsination and were cultured in vitro on six-well plates coated with bovine collagen type I/III (StemCell Technologies, Grenoble, France), as described previously [46,47]. The primary amniocytes in the TIL and TNL groups were respectively collected from fresh fetal membranes: (38–41 WG) and (39–40 WG).

### 4.3. CK2 and RAGE Immunofluorescence Staining

Immunohistochemistry experiments were performed on 8-µm FM cryosections or cultures of primary amniocytes. The samples were fixed in paraformaldehyde (in 4% phosphate-buffered saline [PBS]); blocked with PBS-1X, Triton 0.1%, and SVF 10%; and incubated overnight with a polyclonal rabbit anti-CK2α antibody (generously given by Dr Odile Filhol, UGA-CEA-INSERM U1292, Grenoble, France), polyclonal rabbit anti-CK2α′ antibody (Bethyl Laboratories, Montgomery, TX, USA), polyclonal rabbit anti-CK2β antibody (Thermo Fisher Scientific, Asnières-sur-Seine, France) diluted at 1:100, or polyclonal rabbit anti-RAGE antibody (Abcam, Paris, France) diluted at 1:200. A secondary antibody (donkey anti-mouse antibody coupled with Alexa 488 IgG [H+L] (Thermo Fisher Scientific) diluted at 1:1000 was incubated for 2 h on slides. After Hoechst nuclear staining (Sigma-Aldrich, Saint-Quentin-Fallavier, France), the samples were examined under a LSM800 confocal microscope (Zeiss, Rueil Malmaison, France) with the Zen 2.3 microscopy software application (blue edition). As negative controls, samples were incubated without a primary antibody.

### 4.4. Quantitative RT-PCR

Total RNA was isolated from the amnion, choriodecidua, and primary amniocyte cells using the NucleoSpin RNA kit (Macherey-Nagel, Hoerdt, France), in accordance with the manufacturer’s protocol. After quantification with a DS-11Fx spectrophotometer (DeNovix, Wilmington, DE, USA), cDNA was synthesized from 1 μg of RNA using a SuperScript™ IV First-Strand Synthesis System for RT-PCR (Invitrogen, Carlsbad, CA, USA). Quantitative RT-PCRs were performed with LightCycler 480 (Roche, Meylan, France) using a Power SYBR Green Master Mix (Roche) and specific primers (as described in Table 1). Transcripts were quantified using a standard curve method in the LightCycler 480 Software release 1.5.1.62. The ratio of the gene interest transcript to the geometric mean of two housekeeping genes (RPLP0 and RPS17) was determined. The results were obtained from at least six independent experiments, and all the steps followed the guidelines in the Minimum Information for Publication of Quantitative Real-Time PCR Experiments (MIQE) [48].

### 4.5. Western Blot Analysis

The amnion and choriodecidua samples were homogenized as described in a previous report [12]. Primary amniocyte cells were lysed in a radioimmunoprecipitation assay buffer (20 mM Tris-HCl at pH 7.5, 150 mM NaCl, 1% Nonidet P40, 0.5% sodium deoxycholate, 1 mM EDTA, 0.1% SDS, 1× complete protease inhibitor cocktail [Roche]) for 30 min at 4 °C. The protein concentration of the supernatant was determined using a bicinchoninic acid assay kit (Pierce). Owing to the weak concentration of amniocyte samples, they were mixed in two pools, each bringing together four different amniocyte preparations. Thirty micrograms of denatured protein was subjected to a Western blot analysis after 4–15% Mini-PROTEAN TGX StainFree gel electrophoresis (Bio-Rad, Marnes-la-Coquette, France), followed by probing proteins against rabbit polyclonal CK2α and mouse monoclonal CK2β antibodies, which were generously given by Dr Odile Filhol (UGA-CEA-INSERM U1292, Grenoble, France) and diluted at 1:1000. The rabbit polyclonal CK2α′ (Bethyl Laboratories) and the mouse monoclonal RAGE (Santa-Cruz Biotechnology, Santa Cruz, CA, USA) antibodies were diluted at 1:2000 and 1:400, respectively. The signal was detected with a peroxidase-labeled anti-mouse antibody at 1:10,000 (Abliance, Compiègne, France) and visualized with Clarity Western ECL substrate (Bio-Rad) using the ChemiDoc MP Imaging System and Image Lab software (version 6.0.1. build 34; Bio-Rad). The relative CK2 subunit ratio was expressed in the function of the total protein loaded per well.

### 4.6. CK2 Activity Assay

The reaction was started by the addition of the sample (3 μL) to the reaction mixture (50 mM Tris-HCl, pH 7.5, 10 mM MgCl_2_, 150 mM NaCl, 20 μM ATP, 150 μM synthetic peptide RRREDEESDDEE as a consensus CK2 peptide substrate [49], and 1 μCi of [γ-^32^P]-ATP/reaction in a final volume of 20 μL). After 5 min at room temperature, the reaction was stopped with 60 μL of 4% cold trichloroacetic acid. After 30 min of precipitation at 4 °C, the reaction mixture was centrifuged, and ^32^P incorporation into the peptide substrate was determined by spotting the supernatant onto phosphocellulose paper disks (Whatman P-81). After three successive 15-min washes in 75 mM cold phosphoric acid, the radioactivity was measured in 5 mL of scintillation liquid (Ultima Gold PerkinElmer, Villebon sur Yvette, France), using a liquid scintillation analyzer (Tri-Carb 2100 TR, Packard BioScience, Meriden, CT, USA). The results are reported as cpm. As a negative control, the CK2 peptide substrate was omitted in the reaction.

### 4.7. Statistical Analyses

The results were analyzed using the PRISM software (version 7; GraphPad Software Inc., La Jolla, CA, USA). Quantitative data are presented as median ± interquartile range. Non-normally distributed data between the two groups were studied using a Mann–Whitney test. To study several independent groups, a Kruskal–Wallis test was performed followed by Dunn’s multiple comparison test. All results were considered significantly different at * *p* < 0.05, ** *p* < 0.01.

## Figures and Tables

**Figure 1 ijms-24-04074-f001:**
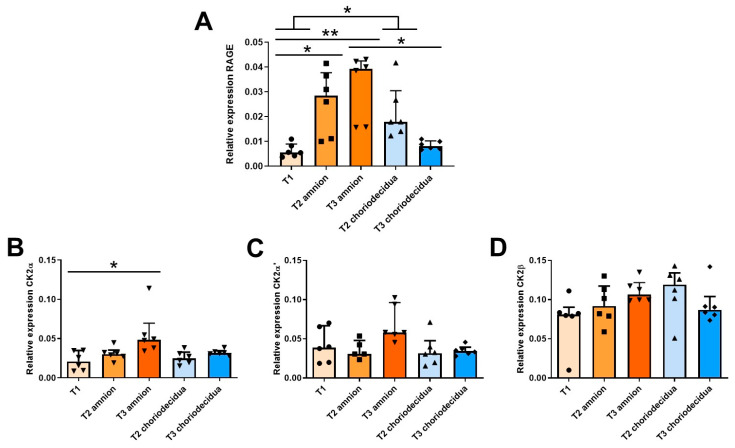
Ontogeny profiling of RAGE and CK2 subunit mRNA expressions in human fetal membranes at T1 (*n* = 6), T2 (*n* = 5–6), and T3 (*n* = 6) of pregnancy. The RAGE (**A**), CK2α (**B**), CK2α′ (**C**), and CK2β (**D**) transcripts were quantified in reverse transcription quantitative polymerase chain reaction (RT-qPCR) experiments. Data are shown as median ± interquartile range. * *p* < 0.05, ** *p* < 0.01.

**Figure 2 ijms-24-04074-f002:**
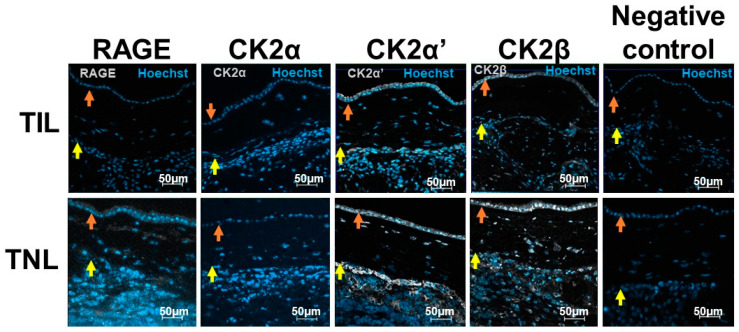
RAGE and CK2α, CK2α′, and CK2β localizations in human fetal membranes at term in spontaneous labor (TIL) or term without labor (TNL) by immunofluorescence. RAGE and CK2 subunits were stained in gray (Alexa 488), and the nucleus was stained in blue with Hoechst. The negative controls were incubated with a primary antibody-free incubation solution. The original magnification is ×200. The orange arrows indicate the amnion, and the yellow arrows indicate the choriodecidua.

**Figure 3 ijms-24-04074-f003:**
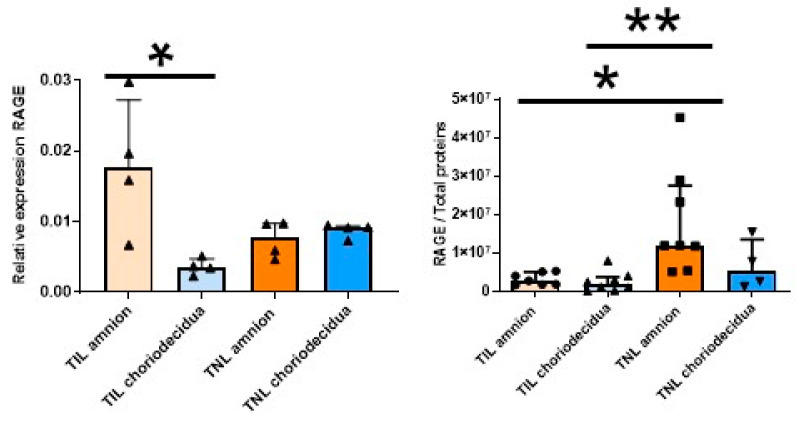
Quantification of the expression of RAGE in human fetal membranes at term with spontaneous labor (TIL) or without labor (TNL). RAGE mRNA (**left**, *n* = 4) and protein expression levels (**right**, *n* = 4–8)) were respectively quantified using RT-qPCR and Western blot assays. Data are shown as median ± interquartile range. * *p* < 0.05, ** *p* < 0.01.

**Figure 4 ijms-24-04074-f004:**
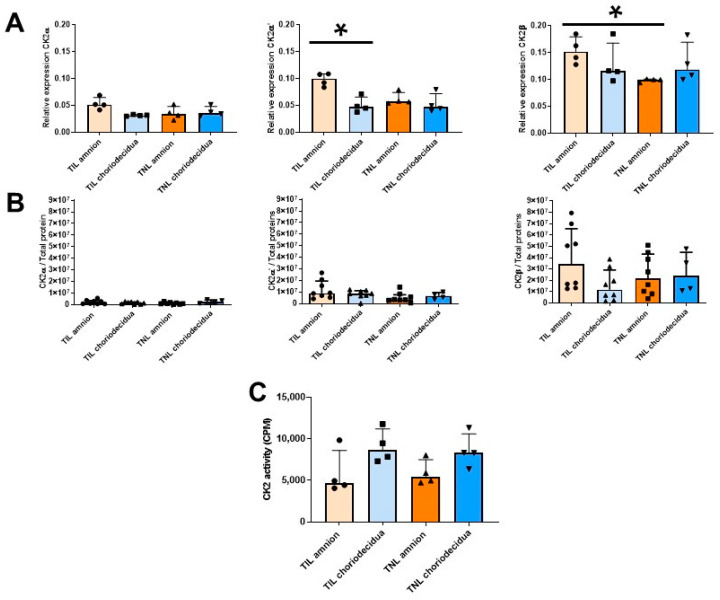
Quantification of the expressions of the CK2 subunits and measurement of the CK2 kinase activity level in human fetal membranes at term with spontaneous labor (TIL) or without labor (TNL). CK2α, CK2α′, and CK2β mRNA expression levels ((**A**), *n* = 4); CK2α, CK2α′, and CK2β protein expression levels ((**B**), *n* = 4–8); and CK2 kinase activity level ((**C**), *n* = 4)) were respectively quantified using RT-qPCR, Western blot assays, and in vitro kinase assays. Data are shown as median ± interquartile range. * *p* < 0.05.

**Figure 5 ijms-24-04074-f005:**
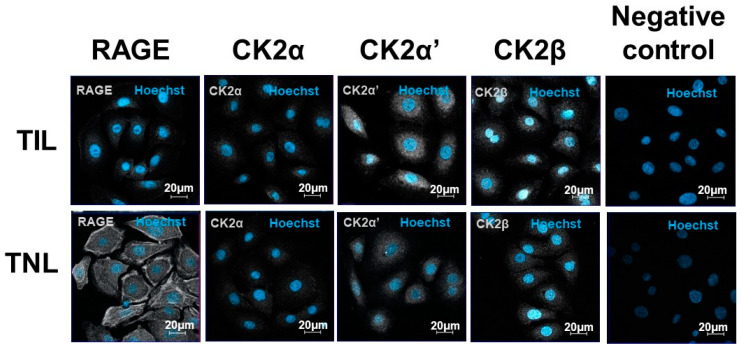
Immunolocalizations of RAGE and the CK2 subunits in the primary human amniotic epithelial cells at term with spontaneous labor (TIL) or without labor (TNL). RAGE and CK2α, CK2α′, CK2β localizations in the human amniotic epithelial cells from the TIL and TNL groups were examined using immunofluorescence. RAGE and the CK2 subunits were stained gray (Alexa 488), and the nucleus stained blue with Hoechst. The negative controls were incubated with a primary antibody-free incubation solution. The original magnification was ×400.

**Figure 6 ijms-24-04074-f006:**
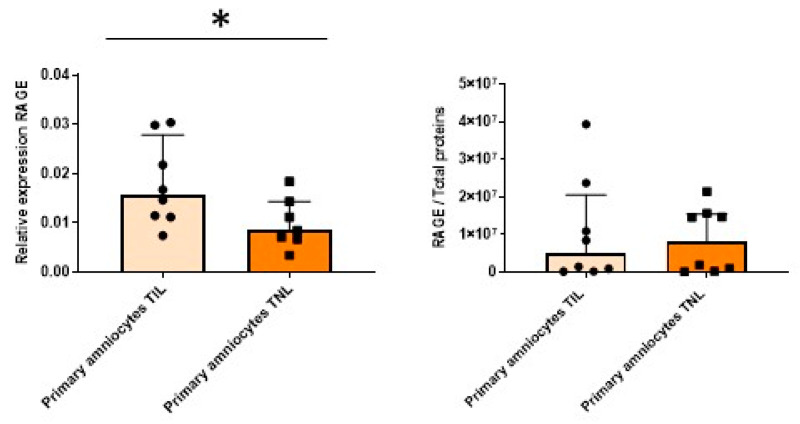
Quantification of the expressions of RAGE in primary amniocytes at term with spontaneous labor (TIL) or without labor (TNL). RAGE mRNA (**left**, *n* = 7–8) and protein expression levels (**right**, *n* = 8) were quantified using RT-qPCR and Western blot assays. Data are shown as median ± interquartile range. * *p* < 0.05.

**Figure 7 ijms-24-04074-f007:**
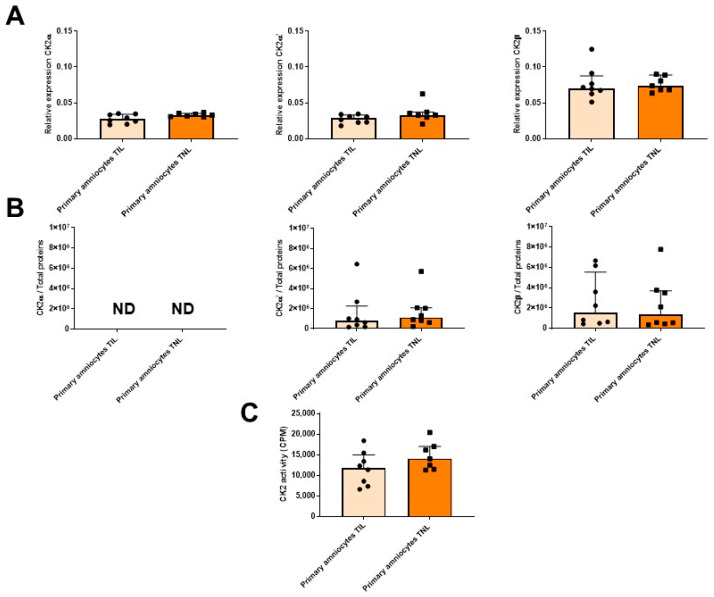
Quantification of the expressions of CK2 subunits, measurement of CK2 kinase activity level in primary amniocytes at term with spontaneous labor (TIL) or without labor (TNL). CK2α, CK2α′, and CK2β mRNA ((**A**), *n* = 7–8), protein expression levels ((**B**), *n* = 8), and CK2 kinase activity level ((**C**), *n* = 7–8) were quantified using RT-qPCR, Western blot assays, and in vitro assays. Data are shown as median ± interquartile range.

**Table 1 ijms-24-04074-t001:** Primer sequences used for the quantitative polymerase chain reaction.

Human Gene	Sequence (5′→3′)	Product Length (bp)	NCBI Reference
hCK2α-S	TGTCCGAGTTGCTTCCCGATACTT	104	NM_177559
hCK2α-A	TTGCCAGCATACAACCCAAACTCC		
hCK2α′-S	AGCCCACCACCGTATATCAAACCT	92	NM_001896
hCK2α′-A	ATGCTTTCTGGGTCGGGAAGAAGT		
hCK2β-S	TTGGACCTGGAGCCTGATGAAGAA	101	NM_001320
hCK2β-A	TAGCGGGCGTGGATCAATCCATAA		
hRAGE-S	TGTGCTGATCCTCCCTGAGA	139	NM_001136.5
hRAGE-A	CGAGGAGGGGCCAACTGCA		
hRPLP0-S	AGGCTTTAGGTATCACCACT	219	NM_053275
hRPLP0-A	GCAGAGTTTCCTCTGTGATA		
hRPS17-S	TGCGAGGAGATCGCCATTATC	170	NM_001021
hRPS17-A	AAGGCTGAGACCTCAGGAAC		

## Data Availability

The data that support the findings of this study are available from the corresponding author upon reasonable request.

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
