# Peer review of "Characterization of RAGE and CK2 Expressions in Human Fetal Membranes"

_ijms, 2023, doi:10.3390/ijms24044074_

Round 1

Reviewer 1 Report (Previous Reviewer 1)

     The aim of this study was to investigate the mRNA and protein expression trend of RAGE gene and CK2α, CK2α and CK2β subunits in amniotic cells and primary amniotic cells after pregnancy, full-term natural delivery (TIL) and non-delivery cesarean section (TNL). This work paves the way for future experiments to regulate RAGE expression by phosphorylation of CK2. It is a job well done. But there are still two problems that need to be modified:

   1. Whose receptor RAGE is in the abstract is not clearly described by the author.

    2. In the abstract, the author did not mention why CK2 was studied and what is the relationship between CK2 and FM rupture, and suggested the author to make some additions.

Author Response

Please see the attachment in the box.

Reviewer 2 Report (New Reviewer)

The study describes the expression of RAGE and CK2 in human FM from different gestational trimesters. This is a very important issue to be studied since alarmins are involved both in normal parturition and preterm birth and a better understanding of the expression profiles of these biomarkers would help us understand and eventually treat important pregnancy-associated pathologies. 

However, I refer my comments below and according to my opinion, further analyses with a different approach should be performed before the study is suitable for publication. 

- Introduction

Well-written and explicative introduction. However, a couple of sentences describing the aim of the study is lacking. In lines 76-81 there is a clear statement of the conclusions of the study but a description of why this study was done would really help the reader to better understand the importance and the conclusions of this study. 

Line 81: the term sterile inflammation should be used more cautiously in my opinion in this study. There is no reference to the definition of the term and no data regarding, amniotic fluid cytokine/interleukine stays, microbiology and histopathology of the FM/placenta in the study confirming the sterile nature of the inflammation in the included cases.

- MM section and results

The period during the study was performed is not referred in the manus. Moreover, the number of pregnancies/women/samples included in the study is not neither reported. 

The term gestational pathologies should not be referred to placental pathologies only. It is important to know if these pregnancies were complicated with preeclampsia, pregnancy-induced hypertension, gestational diabetes, symptoms of preterm birth, drugs used during pregnancy etc. It should be also important to describe the histopathological analyses of the placentas. Was it performed in all cases/trimesters? How? How was the sterile inflammation been confirmed/assumed?

Lines 325-326: "second trimester FM were collected after premature termination of pregnancy without impact om FM". This statement could be further clarified. What exactly authors mean? Is it referred to terminations of pregnancy? which are the reasons in this case?  

Lines 327-329: FM from TIL are cases where labour began previously to rupture of FM or the rupture of FM preceded labour? Which were the reasons for the elective Cesarean sections?

Lines 331-332: the separation of the FM were done after the storage to -80 degrees or after thawing the FM, before analysis?

Line 335: could authors give an explanation of the term "primary amniocyte cells"?

Moreover, in lines 87-88, second trimester (T2) is defined as GW 22-25. Second trimester should be instead GW 13-21 and third trimester (T3), GW 22 to birth. According to the definition of gestational trimesters used by the authors, no real cases of T2 are included in the study. All cases in the study are either T1 or T3. In the results section the study compared T1 vs. T2 and T3. According to the above mentioned definitions of gestational trimesters, T2 and T3 cases should be clustered in one category and compared with T1. However, a better description of the causes of T2 (according to the study) pregnancy termination should be done in order to avoid clustering "normal" term deliveries with pathological ones.  

This is a very important methodological issue to be clarified by the authors before further review of the study. 

Author Response

Please see the attachment in the box.

Round 2

Reviewer 1 Report (Previous Reviewer 1)

As for the present state of the paper, I think there is no need to make any improvement.

Reviewer 2 Report (New Reviewer)

I would like to thank the authors for their prompt answers to my recommendations that could make the understanding of the work easier.

However, I would suggest that some of the information provided in their answers would make the reading easier to understand and to avoid misunderstandings if they were also included in the manuscript (MS). For example: 

1. answer to Rek. 2 and 5: it is important to highlight in the MS that the definition of sterile inflammation used in this work is according to the French clinical guidelines and it doesn't include AF analysis. 

2. answers to Rek. 6, 7 and 9: these informations should be included in the new version of the MS. 

3. answer to Rek. 10: in order to avoid misunderstanding and to easier understand the methodology of the work, I suggest to the authors to place the information about the trimesters of pregnancy when FM were collected in the MM section instead of results section of the MS. 

Author Response

In order to have an easier reading of the article, the authors followed the recommendations of the reviewer2 and included all his comments in the manuscript (points 1, 2 and 3).

Round 3

Reviewer 2 Report (New Reviewer)

I would like to thank the authors for their work with the article. Good luck with  their studies in the field. 

This manuscript is a resubmission of an earlier submission. The following is a list of the peer review reports and author responses from that submission.

Round 1

Reviewer 1 Report

1. In line 87 of the manuscript, based on the understanding of the context, it should be necessary to add "(T3)" after the third trimester, suggesting the author to revise it.

2. In FIG. 3 and FIG. 5, the manuscript both mentioned the use of RT-qPCR, WB and in vitro kinase to test the activity level of CK2 kinase, but the author only put in one diagram. I am puzzled why not three diagrams, please explain in the paper.

3. In Figure 2 and Figure 4, the font size of the ruler of the image enlargement ratio is different, so it is suggested that the author adjust it to a uniform size.

4. In FIG. 1, FIG. 3 and FIG. 5, the serial numbers of A, B, C and D in the upper left corner of the picture are also different in size, so it is suggested that the author adjust them together.

Author Response

We thank the reviewer for the positive and constructive comments regarding the article.

Recommendation 1. In line 87 of the manuscript, based on the understanding of the context, it should be necessary to add "(T3)" after the third trimester, suggesting the author to revise it.

Answer to recommendation 1: We agree and decided to add T3 in line 87.

Recommendation 2. In FIG. 3 and FIG. 5, the manuscript both mentioned the use of RT-qPCR, WB and in vitro kinase to test the activity level of CK2 kinase, but the author only put in one diagram. I am puzzled why not three diagrams, please explain in the paper.

Answer to recommendation 2: Making a single figure was to gather biochemical and molecular results for RAGE and the different isoforms of CK2. As reading the figures 3 and 5 seems confusing to the reviewer 1, we decided to split them into two parts: one on RAGE and the second on CK2:

Figure3 --> Figures 3 and 4

Figure 5 --> Figures 6 and 7.

Recommendation 3. In Figure 2 and Figure 4, the font size of the ruler of the image enlargement ratio is different, so it is suggested that the author adjust it to a uniform size.

Answer to recommendation 3: We agree with the reviewer’s comment and decide to homogenize the font size in Figures 2 and 4.

Recommendation 4. In FIG. 1, FIG. 3 and FIG. 5, the serial numbers of A, B, C and D in the upper left corner of the picture are also different in size, so it is suggested that the author adjust them together.

Answer to recommendation 4: Based on this comment, we homogenize the font size of the serial numbers in the different figures.

Reviewer 2 Report

These study is trying to measure the RAGE and CK2 expression level (mRNA and protein) throughout gestation and also compare them at term with spontaneous labor(TIL) and no labor(TNL). Suggestions and concerns are listed as below:

1.     line 88, to be consistent,“(T3)”should be inserted after “third trimester”.

2.     Use consistent format for the axis titles, for example, keep upper case for all the first letter in figures.

3.     In figure 2 and 4 legend, “the CK2 subunits were stained in gray (Alexia 488)”, should it be Alexa488 ? It is also suggested to show the staining in separated blue, green channel and then merged channel in figure 2 and 4. Similar in figure 4, the separate channel will help to quantify and compare the expression intensity for TIL and TNL.

4.     In addition, RAGE and CK2 subunits are recommended to be stained in different colors/channels so that the colocalization or interaction can be clearly verified.

5.     In the second paragraph of section 2.2, the detection efficiency of CK2 subunits antibodies might be different, so it is not proper to compare the relative expression level according to the immunofluorescence staining intensity for different subunits.

6.     In figure 3,  it is suggested to put the groups TIL amnion and TNL amnion together, TIL choriodecidua and TNL choriodecidua together so that the comparison is more clear.

7.     In line 127, “37-41 WG”is not consistent with the information described in section 2.1 for figure 1A, so it might not be proper to compare the expression level in different stage?

8.     In figure 3, 4 and 5, how to quantify RAGE or CK2/total protein using western blot ? Usually, people quantify the expression level normalizing to internal control such as GAPDH, beta-actin etc. in the same western blot membrane.

9.     There are so much inconsistency especially the mRNA and protein levels for this study, the quantification methods should be double checked. Although there might be correlation between RAGE and CK2 from other literature, direct evidences from this study were not found.

Author Response

We thank the reviewer for the positive and constructive comments regarding the article.

Recommendation 1.     line 88, to be consistent “(T3)” should be inserted after “third trimester”.

Answer to recommendation 1: We agree and decided to add T3 in line 88.

Recommendation 2.     Use consistent format for the axis titles, for example, keep upper case for all the first letter in figures.

Answer to recommendation 2: According to the reviewer’s recommendation, we change the format for the axis titles, beginning with an upper case for the first letter in the different figures.

Recommendation 3.     In figure 2 and 4 legend, “the CK2 subunits were stained in gray (Alexia 488)”, should it be Alexa488? It is also suggested to show the staining in separated blue, green channel and then merged channel in figure 2 and 4. Similar in figure 4, the separate channel will help to quantify and compare the expression intensity for TIL and TNL.

Answer to recommendation 3: We agree with the reviewer’s comment and change Alexia in Alexa488 in the text.

We did not choose the presentation mode (grey, blue, merge) with 6 images for each protein (TNL: target protein, Hoechst, merge, TIL: target protein, Hoechst, merge), because the figure would be too complicated to read. However, we propose to create figures as suggested by the reviewer 2 and add it as supplementary materials to be helpful for the future readers (Supplemental figures 1 to 4).

Recommendation 4.     In addition, RAGE and CK2 subunits are recommended to be stained in different colors/channels so that the colocalization or interaction can be clearly verified.

Answer to recommendation 4: The aim of this article is to describe the mRNA and protein expressions of RAGE and CK2. That’s the reason we didn’t use powerful experiments like FRET or PLA (Proximity Ligation Assay) to demonstrate an eventual colocalization or the RAGE-CK2 interaction after co-immunoprecipitation.

Recommendation 5.     In the second paragraph of section 2.2, the detection efficiency of CK2 subunits antibodies might be different, so it is not proper to compare the relative expression level according to the immunofluorescence staining intensity for different subunits.

Answer to recommendation 5: We agree with the reviewer’s comment and modify the sentence « At term, even if precise quantification using immunofluorescence staining was not adequate, the CK2α, CK2α¢, and CK2β subunit protein expressions exhibited the same pattern as the RAGE expression in FM, but the signal intensities for CK2α¢ and CK2β were higher than that for RAGE. Moreover, the CK2α isoform was weakly expressed compared with CK2α¢ and CK2β » (line 113-117). We propose « At term, even if precise quantification using immunofluorescence staining was not adequate, the CK2α, CK2α¢, and CK2β subunit protein expressions exhibited the same pattern as the RAGE expression in FM with different signal intensities ».

Recommendation 6.     In figure 3, it is suggested to put the groups TIL amnion and TNL amnion together, TIL choriodecidua and TNL choriodecidua together so that the comparison is clearer.

Answer to recommendation 6: The aim of this paper is to study the influence of labor at term on RAGE and CK2 expression in the two layers of fetal membranes. It would be unfortunate to remove this information. As cellular processes are different in case of labor or not labor, it was thus important to study these two conditions in each layer of fetal membranes as clearly stated by other referent teams in Fetal Membranes field (Menon, R. Fetal Inflammatory Response at the Fetomaternal Interface: A Requirement for Labor at Term and Preterm. Immunol Rev 2022, 308, 149–167).

Recommendation 7.     In line 127, “37-41 WG” is not consistent with the information described in section 2.1 for figure 1A, so it might not be proper to compare the expression level in different stage?

Answer to recommendation 7: In the figure 1, we studied the mRNA expression of RAGE and CK2 during the 3 trimesters of the pregnancy, before the term. In human species, the term pregnancy is between 37-41 weeks of gestation (WG). So, we had 6 samples of fetal membranes at T3 before the term (31–36 WG).

In the other figures, we studied the expression of RAGE and CK2 at the time of delivery (after parturition at full term after labor (TIL) or after birth by caesarean section (TNL). We had 4 samples of fetal membranes at term (37-41 WG).

Recommendation 8.     In figure 3, 4 and 5, how to quantify RAGE or CK2/total protein using western blot? Usually, people quantify the expression level normalizing to internal control such as GAPDH, beta-actin etc. in the same western blot membrane.

Answer to recommendation 8: The use of stain-free gel and ChemiDoc MP technologies provides an excellent solution for normalization between the lanes of western blots by total protein transferred. We are totally independent of house-keeping proteins like GAPDH or β-actin whom the expression can change in function of the tissue? For example, the labor is associated with different processes like inflammation with potential modification of the expression of GAPDH and beta-actin. We use this quantification method for several years and published the western blots in several publications:

  • Gross et al; Advanced Glycation End Products and Receptor (RAGE) Promote Wound Healing of Human Corneal Epithelial Cells, Invest Ophthalmol Vis Sci 2020 Mar 9;61(3):14.
  •  
  • Choltus et al; Occurrence of a RAGE-Mediated Inflammatory Response in Human Fetal Membranes, Front Physiol, 2020 Jun 25;11:581.
  • Lavergne et al; Human Amnion Epithelial Cells (AECs) Respond to the FSL-1 Lipopeptide by Engaging the NLRP7 Inflammasome, Front Immunol 2020 Aug 7;11:1645.
  • Choltus et al; Cigarette Smoke Condensate Exposure Induces Receptor for Advanced Glycation End-Products (RAGE)-Dependent Sterile Inflammation in Amniotic Epithelial Cells, Int J Mol Sci 2021 Aug 3;22(15):8345.

  • Antoine et al; Dysregulation of the Amniotic PPARγ Pathway by Phthalates: Modulation of the Anti-Inflammatory Activity of PPARγ in Human Fetal Membranes, Life (Basel), 2022 Apr 6;12(4):544.

Recommendation 9.     There are so much inconsistency especially the mRNA and protein levels for this study, the quantification methods should be double checked. Although there might be correlation between RAGE and CK2 from other literature, direct evidences from this study were not found.

Answer to recommendation 9:

This study is an original work. As far as we know, we didn’t read a direct correlation between RAGE and CK2 expressions in publications or after in silico analyses with the public database STRING (https://string-db.org/). We agree with the comment of the reviewer 2 that we have inconsistency between the mRNA and protein levels. But we performed with enough samples to make non-parametric statistical analyzes, the interquartile are not so large. We hypothesize that these heterogenous results between mRNA and proteins are probably due to different post-transcriptional regulations in case of labor or not.

Round 2

Reviewer 2 Report

1.     In figure 1, please check carefully to unify the format of column titles for “ amnion”, “Amnion”, “Choriodecidua”, “choriodecidua”.

2.     It is possible that the mRNA and protein showed reverse trend because of  existing post-translational modification. However, compare TIL and TNL, the immunofluorescence results are not consistent with the primary cells WB protein expression for amnion , please clarify.

3.     Please include a positive and negative control for CK2 activity detection.
